# Cancer Vaccines: From the State of the Art to the Most Promising Frontiers in the Treatment of Colorectal Cancer

**DOI:** 10.3390/pharmaceutics15071969

**Published:** 2023-07-17

**Authors:** Eleonora Martinis, Carolina Ricci, Caterina Trevisan, Gaia Tomadini, Silvia Tonon

**Affiliations:** Department of Medicine, University of Udine, Piazzale Kolbe 4 Udine, 33100 Udine, Italy

**Keywords:** CRC, cancer vaccine, B-cells, tumor antigens, clinical trials

## Abstract

Colorectal cancer represents 10% of all new cancer cases each year and accounts for almost 10% of all cancer deaths. According to the WHO, by 2040 there will be a 60% increase in colorectal cancer cases. These data highlight the need to explore new therapeutic strategies. Classical interventions include surgical resection, chemotherapy and radiotherapy, which are invasive strategies that have many side effects on the patients and greatly affect their quality of life. A great advance in the treatment of this cancer type, as well as of all the others, could be the development of a vaccination strategy preventing the onset, the progression or the relapse of the pathology. In this review, we summarize the main vaccination strategies that are being studied for the treatment of colorectal cancer (CRC) and finally explore the possibility of using B-cells for the development of a new type of vaccine.

## 1. Introduction

Cancer cells are autologous cells that have acquired abnormal capabilities, which makes it very difficult to find therapies that target only cancer cells while sparing healthy tissues. To date, traditional therapies target both cancer and normal highly proliferating cells (e.g., ovarian follicular cells, intestinal cells, hematopoietic cells), causing much collateral damage. However, in recent years, alternative therapeutic strategies based on enhancing immune responses only toward tumor cells have shown promising results. Named cancer immunotherapies, these innovative strategies have dramatically changed the outcome of several types of cancers, constituting a good alternative for the treatment for metastatic melanoma [1], non-small cell lung cancer [2], cutaneous squamous cell carcinoma [3], urothelial carcinoma [4], refractory Hodgkin’s lymphoma [5], hepatocellular carcinoma [6], gastric carcinoma [7] and triple-negative breast cancer [8]. Several immunotherapies have been or are being developed including immune checkpoint blockade therapy, cytokine therapy, adoptive cellular immunotherapy. However, still high percentages of patients do not benefit from the actual immunotherapeutic protocols. This is particularly true for patients affected by colorectal cancer (CRC), a cancer type whose incidence accounts for 10% of the total worldwide tumor cases, with a high fatality rate (10%) (data source: GLOBOCAN 2020). Indeed, immunotherapies for CRC, mainly consisting of immune checkpoint inhibitors, are limited to patients with microsatellite stability/mismatch repair proficiency cancers [9]. Therefore, the standard approaches for treating the disease are surgery, chemotherapy and radiotherapy, with consequent disadvantages mainly due to non-specificity and cytotoxicity and still many patients succumbing to relapse. Consequently, it is now essential to develop more precise and effective approaches to treat CRC [10]. The latest discoveries have highlighted the possibility of developing anti-cancer vaccines for protective, curative and relapse-preventive purposes [11]. As in many “standard” uses of vaccines, also for CRC many strategies have been explored: peptide-, nucleic acid-, viral vector-, bacterial vector-, yeast vector- and cell-based formulations. In this review, the latest advances in the development of a vaccine against CRC are summarized, and in the last paragraph, we explore the possible use of B lymphocytes for the development of innovative cell-based vaccines. 

## 2. How Do Cancer Vaccines Work?

According to the CDC (Centre for Disease Control and Prevention), a vaccine is “A product that stimulates a person’s immune system to produce immunity to a specific disease, protecting the person from that disease”.

Similar to vaccines against infectious diseases, cancer vaccines represent new therapeutic tools in the fight against tumors, designed to boost the capacity of an individual’s immune system to recognize and react against specific antigens of the cancer cells. 

In the context of cancer vaccines, there is a first important subdivision to be made: preventive cancer vaccines and therapeutic cancer vaccines. 

Preventive cancer vaccines can be administered prior to the onset of a tumor or in the pre-malignant state, before the establishment of an immunosuppressive tumor microenvironment (TME), thus inhibiting the tumor further progression [12]. They can be developed to elicit an adaptive immune response against tumor antigens resulting from the accumulation of driver mutations that occur during carcinogenesis. Alternatively, in the case of tumors with an infectious etiology, they can be designed to block infection by the respective causative agents. For example, vaccines against hepatitis B (HBV) and human papillomavirus (HPV) have been developed with this strategy. 

Therapeutic vaccines, on the other hand, trigger an immune response against an existing tumor and against residual cancer cells remaining after other treatments. Similar to preventive vaccines, therapeutic vaccines require the identification of tumor antigens to target.

There are two categories of tumor antigens: tumor-associated antigens (TAAs) and tumor-specific antigens (TSAs). TAAs are proteins present on both normal and tumor cells but usually over-expressed in the latter [5]. Therefore, since TAAs are autologous proteins, they are under the control of central and peripheral tolerance; so, TAA-based vaccines could trigger the elimination of T-cells that recognize those antigens, which represents a limitation of this approach. In addition, because they are also expressed in normal tissues, TAAs present the risk of vaccine-induced autoimmune reactions [13]. 

TSAs, often referred to as neoantigens, are expressed exclusively on cancer cells. They may arise from somatic mutations that lead to non-synonymous amino acid sequences not present in germinal DNA and therefore unique to malignant cells. Additionally, they may result from alternative mRNA splicing events and post-translational modifications. TSA-based vaccines have several advantages over TAA-based vaccines, including greater immunogenicity and the ability to trigger an efficient tumor-specific immune response, with limited “off-target” effects [14]. Moreover, they are not subject to central immune tolerance [15]. The optimal TSAs for vaccine development, in addition to being expressed only in neoplastic cells or preneoplastic lesions and being absent in healthy cells, as mentioned above, should be subject to genetic mutation early in tumor formation, remaining conserved during tumor progression with respect to both tumor type and patients, to allow for a wider use. However, the complexity and instability of tumor genomes, together with the resulting highly variable TMEs, make the formulation of cancer vaccines very complicated. 

## 3. CRC-Targeting Vaccines

As explained earlier, the identification and characterization of tumor antigens is the first, very complex step in vaccine development. This is true for therapeutic vaccines, but even more for preventive vaccines, for which it will be optimal to identify an antigen even before tumor onset. Therefore, to date, most cancer vaccines are therapeutic, including those specific to CRC. Table 1 shows examples of TAAs and TSAs for CRC vaccine development. The type of vaccine through which they are administered is also specified. All the different vaccine types are characterized by specific mechanisms (summarized in Figure 1 and Figure 2) of functioning and offer different advantages but also limitations. The purpose of this review is to describe these relatively new therapeutic approaches that could represent a real new avenue in the therapy against CRC. 

### 3.1. Peptide-Based Vaccines for CRC

The antitumor peptide-based vaccines consist of epitopes from TAAs or TSAs that are capable of eliciting antitumor responses because, as previously explained, are either overexpressed in the tumor relative to normal tissues (TAAs) or result from mutations that cause them to be recognized as non-self antigens (TSAs) [35,36]. In theory, these peptides, once administrated, should be presented by antigen presenting cells (APCs) through MHC molecules, thus inducing specific T cell responses against the tumor (Figure 1). The advantage of these vaccines is based on their tumor specificity and ease of production. In fact, only the minimal immunogenic sequence is needed, which can be obtained synthetically. However, still many limitations prevent them from being a standard option in cancer therapy. Indeed, vaccines based on epitopes from TSAs, while highly immunogenic, take time and cost to be designed. In addition, they lose their effects if the neoantigens from which they are derived mutate further or if antigen loss occurs as a result of immunoediting by the tumor [37]. TAA epitopes, on the other hand, while usable in vaccine formulations with greater versatility and substitutability than neoantigens, can elicit an immune response not limited to the tumor due to their expression in healthy cells as well. 

Moreover, the efficacy of peptide-based vaccines relies on their specific immunogenicity, a property that depends on their nature and the mechanisms of their presentation. The length of the peptide is a determining factor: a shorter peptide does not require antigen processing by APC, thus inducing T-cell anergy; on the contrary, a longer peptide must be further processed to elicit a proper response [38]. Therefore, immunostimulatory adjuvants must be added to the vaccine formulations to prevent the development of immunological tolerance due to poor DC activation, which constitutes a further issue to be addressed. Despite these problems, peptide-based vaccines that have entered clinical trials so far have shown acceptable tolerability. An example is the non-viral prophylactic cancer vaccine against MUC1, a shared tumor antigen expressed on >80% of human cancers, which was administered to healthy individuals at risk for colon cancer [24]. This peptide-based vaccine was capable of inducing a high-affinity polyclonal memory IgG response that was highly tumor-specific and safe due to selection and affinity maturation in the same human host. The IgGs bound hyperglycosylated MUC1 on human cancer cell lines and tumor tissues but showed no reactivity against fully glycosylated MUC1 on normal cells and tissues; in addition, these antibodies are able to induce antibody-dependent cytotoxicity [24]. In a clinical trial, the administration of peptides derived from TOMM34, RNF43 and VEGFR significantly increased the levels of IgG against these antigens; specifically, the production of immunoglobulins against VEGFR correlated with improved overall survival of patients with advanced CRC [26]. Also, HER2 seems to be an emerging biomarker in CRC. It is classified as an oncogene; therefore, it plays roles in cell proliferation, differentiation, migration, as well as in tumorigenesis and tumor progression. In the CRC context, HER2 alterations mostly consist in gene amplification and missense mutations which typically result in HER2 protein overexpression [39]. In the study of Hattori et al. [33], clinical benefit from the combination of personalized peptide vaccination and chemotherapy was reported. Metastatic CRC patients were vaccinated with a maximum of four personal HLA-matched peptides in combination with 5-fluoruracil-based standard chemotherapy. The peptides derived from SART2/3, multidrug resistance-associated protein 3, Her2/neu, cytochrome B, ubiquitin-conjugating enzyme E2 and CEA elicited the strongest immune responses. A clinical trial (NCT01376505) to test safety, as well as to establish the optimal biological dose of the HER2 peptide vaccine is now ongoing [34]. In this study, the patients receive a HER2/neu peptide vaccine emulsified with an n-muramyldipeptide derivative (nor-MDP) as an adjuvant, emulsified in Montanide (ISA 720). Up to now, no serious adverse reactions or dose-limiting toxicities have been observed.

### 3.2. Nucleic Acid-Based Vaccines for CRC

Nucleic-acid based vaccines include DNA- and RNA-based vaccines (Figure 1). The DNA-based vaccines consist of short sequences of circular DNA—plasmids—designed to deliver genes encoding tumor antigens, with or without immunomodulatory molecules [40]. The DNA-based vaccines are designed to pass through the membrane of APCs and to migrate to the nucleus to initiate transcription. Once transcribed and translated, the resulting antigens can be presented by MHC-I molecules, thus eliciting or augmenting the adaptive immune responses against tumor cells bearing such antigens. In particular, APCs can present epitopes to CD8^+^, CD4^+^ T-cells and also B-cells, activating them [41]. A limitation of the DNA-based vaccines relies on their low immunogenicity, with a subsequent poor translation of preclinical data to clinical trials [42]. Despite the urgent need to improve their immunogenicity along with their stability and delivery, at least two different DNA-based CRC vaccines have been tested so far (Table 1). In the MYPHISMO study, a phase 1 clinical trial, patients with advanced CRC are treated with an anti-PD1 antibody in combination with different doses of the TetMyb vaccine [30]. Myb is a transcription factor involved in the differentiation and proliferation of cells, that is usually overexpressed in malignant conditions [43]. In the corresponding vaccine, the cDNA of human *Myb* (with three inactivating mutations), flanked by the T-cell tetanus epitope (*tet*), was cloned with the pVAX1 vector; the results have not been published yet [30]. 

RNA-based vaccines also deliver genetic information encoding tumor antigens, but they do not require transcription. RNA needs only to enter the cytoplasm, rather than the nucleus, which simplifies the vaccine delivery and avoids any oncogenic potential derived from DNA integration into the host genome. Similar to DNA-based vaccines, RNA-based vaccines can induce both a B-cell-mediated antibody response and CD4^+^ and CD8^+^ T-cell reactions to enhance the clearance of malignant cells. Moreover, the overall immunogenicity is greater than that achieved with DNA-based vaccines, and recently, also their structure, stability and delivery have been improved. Different mRNA-based vaccines for many solid tumors are now in clinical trials. They are based on patient-specific neoantigens (NCT05359354, NCT05198752, NCT03313778) or on a mutated form of KRAS (NCT05202561). To our knowledge, no conclusive results have been published yet, although the NCT03313778 study showed promising results for tolerability and safety [44].

### 3.3. Virus-, Bacteria-, and Yeast-Based Vector Vaccines for CRC

The immune system has evolved to efficiently respond to viruses, bacteria and yeasts, with both innate and adaptive mechanisms working in concert to induce a strong and durable response (Figure 1). Similarly, virus-, bacteria- and yeast-based vector vaccines induce robust immune responses, making them extremely powerful. 

Viral vector-based vaccines are based on the engineering of viruses to express tumor antigens (TA) transgenes. In this way, the viral pathogen-associated molecular patterns (PAMPs) trigger specific pattern recognition receptors (PRRs) to boost the activation of APCs against a specific TA. Many types of recombinant viruses can be engineered to infect specifically DCs, which in turn elicit adaptive immune responses by presenting the encoded TA to CD8^+^ and CD4^+^ T-cells, as previously seen for the other vaccine types. An example is the CEA-expressing viral vector AVX701 that was tested in metastatic colorectal cancer patients and was associated with a prolonged overall survival [19].

Similarly, live attenuated bacteria can be engineered to express TA transgenes and used as vectors to elicit specific, durable and robust immune responses against both PAMPs and transgenes [45]. Live attenuated bacterial vector-based vaccines show a greater level of specificity and efficacy than other cancer vaccines due to their ability to induce a wider plethora of immune responses, both humoral and cell-mediated. In addition, several bacterial species have been shown to preferentially target and colonize solid tumors due to the inherent characteristics of the TME, including the presence of hypoxic areas—the reason why anaerobic bacteria are usually employed—and chemo-attracting compounds such as aspartate, serine, citrate, ribose or galactose produced by tumor cells. Moreover, aberrant neo-angiogenesis occurring during cancer progression results in impaired endothelial linings which allow bacteria to escape from the circulation and lodge within the tumor mass [46,47,48]. In these respect, and as previously mentioned, live attenuated bacteria represent an attractive platform for tumor-targeted therapy. However, to date, only a limited number of preclinical studies have reached clinical trials. Most of them are based on Listeria monocytogenes (Lm), engineered to release neoantigens of different tumors including CRC [49,50]. 

Yeast vector-based vaccines have also been demonstrated to effectively stimulate the activation, maturation and proliferation of effector T-cells against several mutated or overexpressed tumor neoantigens [51]. Unicellular yeasts, including Saccharomyces cerevisiae (*S. cerevisiae*) and Pichia pastoris (*P. pastoris*), are the most widely used vectors in the development of these vaccines, as they successfully display on their surface neoantigens from different tumor types, boosting robust and durable responses [52,53]. Moreover, they are safe, given their non-pathogenic nature, and do not require adjuvants. Finally, the cost-effectiveness of the manufacturing processes offers further hope for the use of these platforms as a common practice in cancer therapy [54,55,56]. 

### 3.4. Dendritic Cell (DC)-Based Vaccines for CRC

DCs are extremely interesting in the field of cancer vaccines, given their ability to acquire and present antigens through a variety of mechanisms, such as cross-presentation, cross-dressing, antigen transfer, and MHC-I- and II-restricted presentation. Through these strategies, they are decisive in activating CD8^+^ T-cells and priming Th1 cell types, such as naive CD4^+^ T-cells, towards antitumor activity [57].

The most commonly used preparation for DC-based vaccines involves the reinfusion of ex vivo-derived DCs after they have been pulsed in vitro with TAs (antigen-loaded DCs) or with tumor cell lysates (whole tumor-loaded DCs) and stimulated with a defined maturation cocktail containing mainly pro-inflammatory cytokines (Figure 2). In this way, DCs acquire the competence to react against the antigen and, when reinfused into the patient, exert their action against tumor cells carrying that antigen.

Examples of these vaccines are reported below:DCs loaded with autologous tumor lysate [58]. DCs treated with tumor lysates derived from needle-core biopsies were injected in patients with resectable metastatic colon cancer. This therapeutic vaccine resulted in an improved disease-free survival. NCT01348256.DCs modified to express tumor antigens [25]. Dendritic cells engineered with the fowlpox virus encoding CEA and MUC1 and costimulatory molecules. The study aimed to compare DCs and poxvector vaccines against CEA and MUC1, reaching the conclusion that both had similar activity, with superior survival of the vaccinated patients compared with the contemporary unvaccinated group. NCT00103142.DCs pulsed with CEA peptide [17]. Ten patients were vaccinated intradermally and intravenously with CEA peptide-pulsed mature DCs three times prior to resection of liver metastases. High numbers of CEA-specific T-cells were detected in post-treatment DTH (delayed-type hypersensitivity) biopsies in 7 out of 10 patients, which produced high amounts of IFNγ upon stimulation.

## 4. Use of Nanotechnologies to Improve CRC Vaccines Efficacy 

Beyond their specific characteristics, the various types of vaccines possess the common goal of mobilizing the immune system so that it can effectively recognize and eliminate the tumor. The first and critical step is the efficient delivery of the vaccines to APCs. In this regard, nanotechnology is now one of the most promising tools to address this challenge. Nanotechnologies consist of a variety of innovative nanomaterials with attractive properties for cancer vaccine design: by conjugating TAAs or TSAs to nanomaterials or encapsulating them, they significantly improve antigen delivery to the lymph nodes and evoke stronger immune responses. Moreover, by simply manipulating certain physical properties of these nanomaterials, including shape, size and charge, it is possible to control and improve some characteristics of the vaccine including its ability to cross biological barriers or APC targeting, thereby increasing their immunogenicity [59,60,61,62]. Several nanomaterials have been explored in recent years, examples of which are given below.

Lipid-based nanoparticles (LNPs) are lipid vesicles formed from self-assembled phospholipids that exhibit low toxicity and high biocompatibility [63]. As mentioned above, their shape, size and charge can be adjusted to optimize their efficacy in antigen delivery. An additional advantage is their ability to co-deliver multiple antigens and adjuvants. So far, they have been successfully used in nucleic acid-based vaccine formulations because of their ability to protect nucleic acid molecules from nucleases. It is worth mentioning the two COVID-19 vaccines mRNA-1273 [64] and mRNA-BNT162b2 [65], which are based on LNPs technology. 

To date, several other LNPs–mRNA-based vaccines are under study or in clinical trial for the prevention and treatment of viral infections, genetic diseases and cancer. In 2020, for example, with the purpose of finding innovative strategies for the treatment of CRC, Lei and co-workers created a liposome-based system to deliver the mRNA of IL-15, a cytokine with immuno-stimulatory capacity and antitumor potential [66]. They showed that this preparation could be successfully delivered into C26 cells and that its local or systemic administration to murine C26 colon cancer models inhibited metastasis with great efficacy and safety. Another example is provided by the mRNA-4157 nanovaccine, encoding multiple neoantigens to induce specific T-cell anti-tumor responses. More in detail, this nanovaccine is based on LNPs encapsulating mRNAs of human OX40L, IL-23 and IL-36γ. A phase I study is underway to evaluate its safety, tolerability and immunogenicity and is being tested alone in patients with resected solid tumors and in combination with pembrolizumab in patients with unresectable solid tumors (NCT03739931) [67].

Protein nanoparticles are other examples of natural nanomaterials characterized by biocompatibility and biodegradability [68]. Among them, virus-like particles (VLPs) are complexes of viral proteins to which antigens can be chemically or genetically coupled. In vaccine formulations, they function as adjuvants, as they can be easily recognized and efficiently incorporated by APCs, enhancing the immune response [69].

VLPs-based vaccines against tumors can be used in combination with chemotherapy, radiotherapy and immunotherapy. To date, VLPs-based vaccines are available against human papilloma virus (HPV) and hepatitis virus. In addition to exogenous VLPs, there are also endogenous self-assembled proteins, also called caged proteins, that can be exploited for antigen delivery due to their highly organized structure [70]. One example is ferritin, an endogenous protein that has been shown to passively target the lymph nodes, eliciting a robust immune response. In vaccine development, antigens can be genetically modified to form subunits of a protein or can be incorporated into its structure to be recognized and processed by APCs [71].

Polymeric nanoparticles are colloidal systems characterized by stable structures that allow the loading of antigens, which can be incorporated into the structure core or exposed on its surface. Due to their high immunogenicity, they function as self-adjuvants, enhancing phagocytosis or endocytosis by APCs [72,73]. They can be derived from natural sources or be synthetic. Natural polymeric nanoparticles, such as chitosan and dextran, are biocompatible and biodegradable. Synthetic polymeric nanoparticles, such as polylactide (PLA) and poly(lactide-co-glycolide) (PLGA), can be more easily reproduced and controlled in their composition. 

Finally, inorganic materials are another category of nanomaterials that are being studied for drug or antigen delivery in vaccine formulations. They include gold and silica nanoparticles. Gold nanoparticles (GNPs) have a high ability to load antigens and an intrinsic capacity to induce the production of pro-inflammatory cytokines, stimulating the immune responses. Similarly, silica nanoparticles can load different antigens and adjuvants within their structure or on their surface, enhancing lymph node targeting and APC uptake [74,75]. 

Currently, nanotechnology is also increasingly being considered for the treatment of CRC, and several advances have been made in recent years to improve the existing standard methods of detection and therapy by exploiting these innovative nanomaterials. In particular, several nanotechnologies are being used or are now in clinical trials for precision diagnosis and drug delivery for CRC patients [76], opening the possibility of their use in the development of CRC nanovaccines as well.

## 5. CRC-Targeting Vaccines under Clinical Trial

To give an idea of the state of the art of clinical trials on cancer vaccines for CRC, we navigated the NIH database ClinicalTrials.gov (https://clinicaltrials.gov/ct2/home (accessed on 20 January 2023)). By using the keywords “CRC, vaccine, Colo-rectal cancer, Vaccination, Colorectal Carcinoma, Adult, Older Adult”, 87 studies were found, distributed as summarized in Table 2. Of note, none of the registered studies have yet reached Phase 4, and only four trials reached Phase 3, but their status is defined as terminated, withdrawn, completed or unknown. Therefore, it is not a speculation saying that there is an urgent need to increase research in this field. In the next paragraph, the current knowledge on the use of B-cells as a new strategy for the development of vaccines is presented.

## 6. B-Cell Vaccines: A Possible Route for Cell-Based Vaccines?

B-cells are mainly known for their ability to produce antibodies, but besides this exclusive function, they can boost the immune response also through other mechanisms, for example, as antigen-presenting cells. Of course, the ability of B-cells to present antigens is usually linked to their capacity of antigen uptake, internalization and processing that depends on surface immunoglobulins specific for that antigen. However, in 1997, Schultze et al. [77] showed that B-cells isolated from peripheral blood and properly stimulated through the CD40 signaling pathway are a source of efficient APC for ex vivo antigen-specific T-cell expansion. Also, Lapointe et al. showed that after CD40 stimulation, antigen-unspecific B-cells isolated from healthy donors and pulsed with lysates prepared from melanoma cells are able to expand specific T-cells of melanoma patients [78]. Interestingly, it was also shown that pulsed CD40-activated B-cells are also able to efficiently process and present antigens also to naïve CD4^+^ T-cells [79]. Compared with DCs, they are more abundant in peripheral blood, can be more easily isolated and expanded and have a higher migration ability to secondary lymphoid organs, when properly activated. All these features make them suitable candidates for developing alternative cell-based vaccines. 

In their work, Ren et al. demonstrated that naïve B-cells activated via CD40 or TLR and loaded with tumor-derived autophagosomes selectively capturing TSAs could lead to tumor regression in E.G7 murine thymoma models [80]. Li et al. investigated the therapeutic efficacy of in vivo tumor-primed and ex vivo activated B-cells to mediate breast cancer regression: in murine 4TI breast cancer models, the adoptive transfer of effector B-cells from tumor-draining lymph nodes (TDLN) further activated with LPS and anti-CD40 mAb resulted in increased IgG secretion, inhibition of spontaneous metastases to the lung and tumor-specific T cell immunity. They also showed that the combined transfer of activated TDLN T- and B-cells led to tumor regression [81]. 

At the beginning of 2021, also Oxley et al. demonstrated that a proper stimulation of B-cells using tumor cell lysates as priming antigens could represent an effective strategy for the development of B-cell-based cellular vaccines (Bvac) against tumors. In this work, highly purified resting B-cells were isolated from C57BL/6 mouse splenocytes and stimulated with R848 (TLR7 agonist), *E. coli* LPS and an anti-CD40 mAb. They were then exposed to either B16F1 tumor lysates (tumor lysate-Bvac) or a cocktail of two known melanoma peptides (tumor peptides Bvac) and transferred into naïve C57BL/6 mice later challenged with a subcutaneous injection of B16F1 cells. Thus, conditioned B-cells upregulated antigen presentation molecules and were able to stimulate CD4^+^ and CD8^+^ T-cell activation and to induce T-cell migration in vitro. Mice vaccinated with tumor lysate–Bvac showed the greatest improved overall survival, a reduction in tumor size and an increased time to tumor appearance compared to those injected with tumor peptides–Bvac or with unstimulated naïve B-cells, used as controls [82]. Between 2018 and 2019, a phase I clinical trial of an immunotherapeutic vaccine based on B-cells and monocytes as antigen-presenting cells (BVAC-B) was underway in HER2-positive advanced gastric cancer patients. The B-cells and monocytes were transfected with a recombinant HER2/neu gene and loaded with alpha-galactosyl ceramide, which is a natural killer T-cell ligand. In mouse models, preclinical data revealed that the vaccine exerted a promising anti-tumor activity, since it elicited several immune responses against HER2/neu-positive tumor cells. In few patients, BVAC-B (NCT03425773) vaccination resulted in the activation of different immune populations such as natural killer T-cells, natural killer cells, HER2/neu-specific T-cells and in the release of a specific antibody repertoire [83]. These are encouraging preliminary data, since B-cells are abundant in the peripheral blood, and tumor lysates can be easily obtained from biopsies. 

## 7. Conclusions and Future Perspectives 

The Babylonian code, the Eberus papyrus, the Edwin Smith papyrus, thousands of years B.C., already described malignant diseases and their treatments [64]. However, research to fight tumors is still far from a resolution. Indeed, despite the great successes achieved in recent decades with preventive measures, early diagnostics and therapy, malignant tumors continue to be among the leading causes of death and deterioration in the quality of life. In particular, colorectal cancer is the third deadliest cancer in the world, accounting for an estimated 10% of total cancer-related deaths in 2020 [65]. Indeed, regardless of the availability of various therapies, surgery, radiotherapy and chemotherapy, the survival rates for CRC patients remain very poor. Although the introduction of immunotherapy for CRC has shown remarkable results for a subset of patients with mismatch repair-deficient mutations or microsatellite instability in their tumors, the great majority of patients do not respond to such therapies. In this scenario, cancer vaccines may represent a novel possibility to increase patient survival rate and to prevent the increase in CRC incidence. A projection by the International Agency for Research on Cancer of the WHO estimates an increase in new cases of CRC worldwide by 2040 of 60% for colon cancer and 50% for rectal cancer. To date, with respect to the incidence of the disease, clinical trials are very few, and unfortunately, most of them are only in the preliminary phases, with no significant results in terms of increased overall survival. From a biological and immunological point of view, we can hypothesize some explanations for these poor results. 

The probability to develop a good vaccine against a tumor depends primarily on the availability of neoantigens that are suitable for MHC presentation and are immunogenic. Neoantigens presence strictly correlates with the tumor mutational burden, a characteristic that varies from tumor to tumor, both within the same tumor type and between cancers of different origins. Compared to other cancer types, CRC presents a lower mutational burden, with the highest number of mutated proteins in the group of patients with microsatellite instability high (MSI-H), which represents only a minor part of CRC cases [84]. Therefore, the absence of a unique TSA for CRC combined with its modest mutational burden can represent a limitation in the development of a vaccine. Another critical point is the need to target antigens that are not shared with healthy structures, as in the case of TAAs, thus avoiding the risk of autoimmune reaction. 

When dealing with a vaccine directed against a pathogen, the main challenge is the identification of the correct antigen for the induction of a strong and protective immune response. This is one of the greatest advantages in the case of infectious diseases: everyone is infected by the same pathogen; so, the only variable affecting vaccine efficacy will be the diversity in the immune system between individuals. A great variability in the immune response derives from the diversity of HLA alleles. The human MHC-I genes are highly polymorphic, especially in the peptide-binding region: each different HLA molecule binds a peptide with variable strength. For example, HLA-B58 is predicted to bind more efficiently the peptide derived from BRAF mutation V600E (a mutation present in some CRC patients), compared to those from other alleles [84]. Going beyond tumor antigens and HLA variability, another factor that must be taken into consideration for the success of a vaccine is the tumor microenvironment (TME). The main target of cancer vaccines is to stimulate cytotoxic CD8^+^ T-cells; however, a great role can be played also by CD4^+^ effector T-cells, since they are involved in the induction and maintenance of immune memory. This mechanism can be negatively influenced by immunosuppressive cells belonging to the TME; for example, myeloid-derived suppressor cells that are increased in CRC primary tissues can inhibit T-cell proliferation [85]. Of note, MSI-H CRC patients usually have a high percentage of active infiltrating lymphocytes, while patients with stable microsatellite (accounting for 95% of the cases) are often characterized by an “immune desert”, preventing the development of an anti-tumor response [86]. 

All these limitations should be addressed and overcome by new technologies; mRNA vaccines, for example, can be employed to target different antigens, as was done for SARS-CoV-2 variants [87]. In addition to mRNA-based vaccines, preclinical and clinical phase I and 2 data from recent years support the tolerability and efficacy of B-cell-based vaccines [83], possibly combined with other therapeutic strategies. Due to the absence of toxicity, minimal costs, and improved outcomes, B-cell-based vaccines may thus provide additional advantages in the treatment of patients with different types of cancer, including CRC. The use of B-cells as APCs is promising, but some points still require further investigation. For example, the delivery of antigens to optimize their loading is still a challenge that needs to be addressed. To date, various possibilities have been explored. One is the isolation and expansion of B-cells with a specific receptor for tumor antigens from biopsies or peripheral blood. Hypothetically, this strategy will generate a highly specific immune response but has some technical limitations (e.g., the tumor-infiltrating B-cells isolation yield), and more importantly, it is necessary to know the tumor antigen. To overcome these obstacles, an interesting approach was proposed by Szeto et al. [88], who explored the possibility of squeezing B-cells through a microfluidic device to create transient pores on the plasma membrane and allow the delivery of proteins from the surrounding medium into the cells. Their approach led to the MHC-I presentation of peptides derived from whole proteins. This strategy could be employed to deliver whole tumor cell lysates to create personalized vaccines. However, even this strategy has some limitations that should be considered, including the absence of antigen presentation by MHC-II molecules and the resulting lack of help generation by CD4 T-cells, which are particularly important for the development of a CD8^+^ T-cell response. Another strategy has been explored in a very recent paper by Garcìa-Ferrares et al. The authors studied the cross-presentation ability of B-cells and their capacity to activate CD8^+^ T-cells after trans-phagocytosis of bacteria from previously infected dendritic cells. They translated this mechanism to a tumor setting, showing that the B-cells capturing bacteria expressing tumor antigens can be used as a therapy against cancer. In their murine melanoma model, the treatment with “instructed” B-cells led to a reduction in the tumor size, highlighting the potential use of this strategy as a new therapy [89]. 

Collectively, differences in antigens, HLA and TME from one patient to another greatly increase the complexity level of the challenge and make difficult the development of a universal vaccine for CRC. Therefore, more clinical trials exploring new technologies are needed. In this view, B lymphocytes, as a source of APCs, and nanotechnologies for a more efficient delivery may be possible solutions, keeping in mind, however, that the identification of the target antigen remains the main concern in vaccine development. Nonetheless, it is hoped that in the next few years, advances in sequencing technologies, bioinformatics and, perhaps, artificial intelligence and machine learning will make it possible to overcome these limitations and easily identify patient-specific antigens and HLA specificity in the perspective of personalized medicine.

## Figures and Tables

**Figure 1 pharmaceutics-15-01969-f001:**
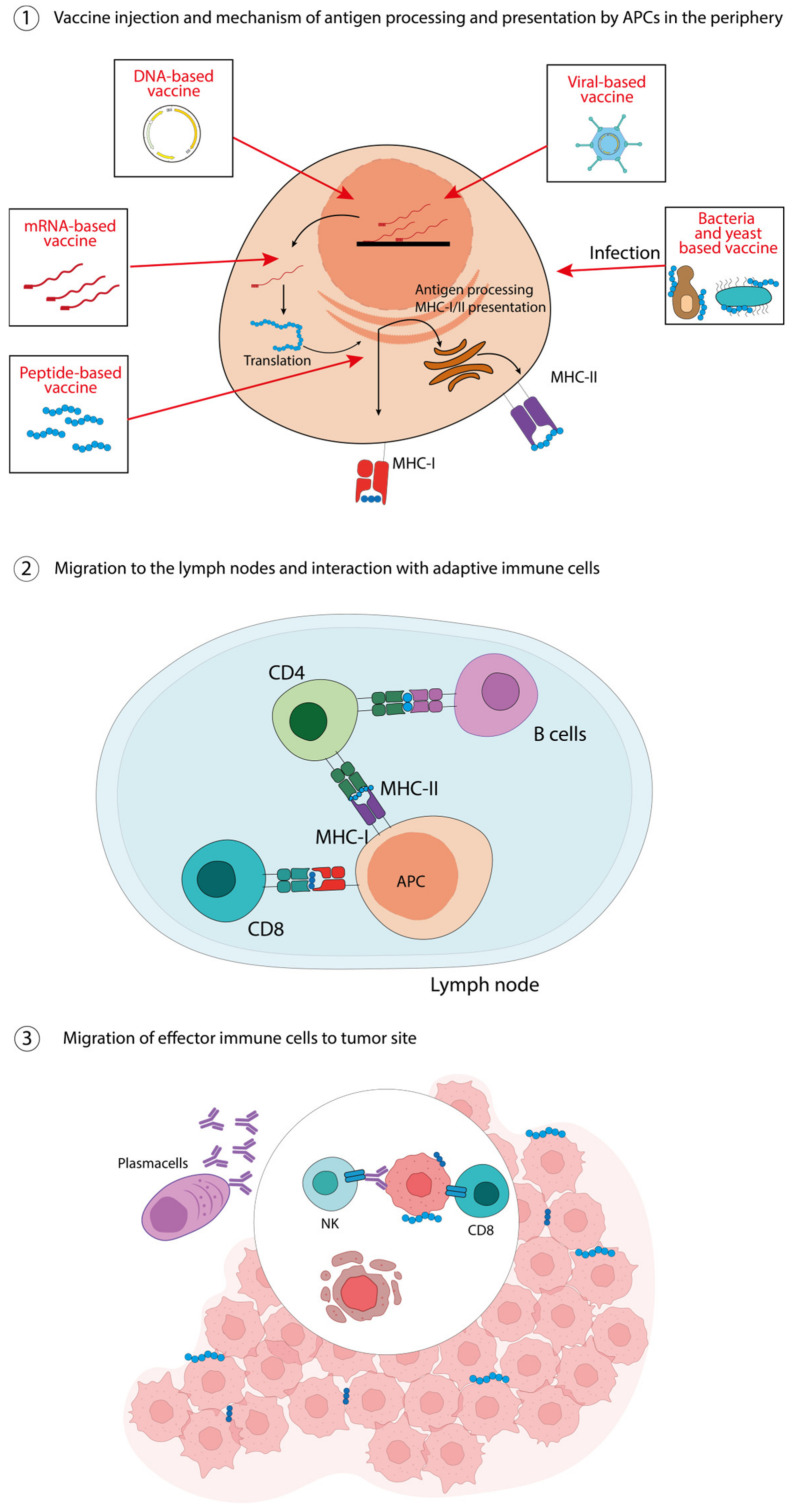
Peptide-, nucleic acid-, virus-, bacteria-, yeast-based vaccines. ① The injected vaccines are taken in by APCs and, according to their nature, are differently processed for their presentation on MHC-I and MHC-II molecules. ② Antigen-loaded APCs migrate to the lymph nodes to activate immune cells, i.e., CD8 T-cells through MHC-I molecules and CD4 T-cells trough MHC-II molecules, which in turn can activate cognate B-cells. ③ Effector cells migrate to the tumor site, the antibodies produced by plasma cells can enhance antibody-dependent cellular cytotoxicity (ADCC), while cytotoxic T lymphocytes can directly kill tumor cells.

**Figure 2 pharmaceutics-15-01969-f002:**
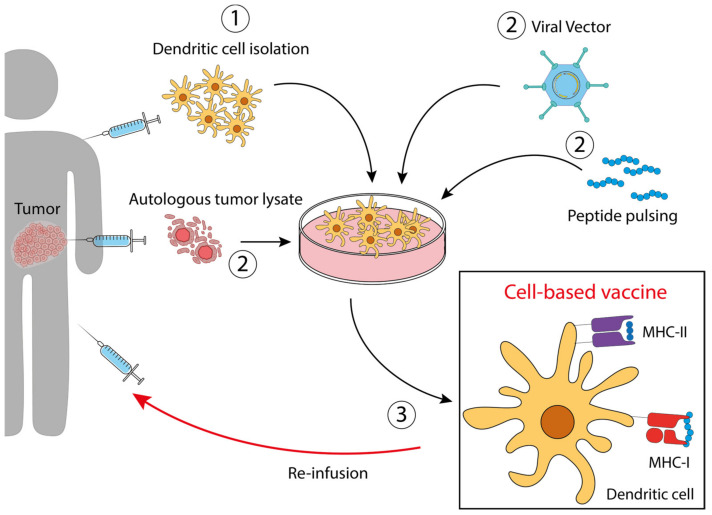
Dendritic cell-based vaccines. ① Dendritic cells precursors are isolated from the patient and expanded ex vivo. ② According to the specific vaccine preparation, DC are treated with an autologous tumor lysate, engineered with a viral vector, or pulsed with a peptide derived from the specific tumor antigen, selected in order to expose the correct antigen on MHC molecules. ③ After treatment and expansion, the antigen-loaded DCs are re-infused into the patient.

**Table 1 pharmaceutics-15-01969-t001:** Example of vaccines for CRC and their related antigens.

Antigen	TAAs or TSAs	Expression in Tumor	Vaccine	Ref.
CEA (carcinoembryonic antigen)	TAA	Increased expression of CEA is associated with adenoma carcinoma, mostly CRC	mRNA, DC-loaded cells, DNA vaccine, viral vector	[16,17,18,19]
RAS	TSA	Mutated in 50% of CRC patients	Peptide, mRNA	[20,21,22]
MUC1 (mucin-1)	TAA	Overexpressed and hypoglycosylated	Peptide, DC-based	[23,24,25]
RNF43 (ring finger protein 43) TOMM34 (translocase of the outer mitochondrial membrane 34) VEGFR (vascular endothelial growth factor receptor)	TAA	CTL-inducing peptide	Peptide	[26]
SART3 (squamous cell carcinoma antigen recognized by T cell 3)	TAA	Overexpressed in the majority of colorectal cancers	Peptide	[27]
β-hCG (beta-human chorionic gonadotropin)	TAA	Expressed at the invasive front of CRC and correlated with poor prognosis	Peptide	[28]
Survivin-2B	TAA	Overexpressed on both cancer and endothelial cells of the tumor vasculature also in CRC	Peptide	[29]
MYB	TAA	Transcription factor that is overexpressed in CRC	Plasmid DNA	[30]
5T4 glycoprotein	TAA	Overexpressed in adenocarcinomas, included CRC	Peptide, Viral vector	[31,32]
Her2	TAA	Gene alterations in CRC include amplification and missense mutations, often mirrored by protein overexpression	Peptide	[33,34]

**Table 2 pharmaceutics-15-01969-t002:** List of clinical trials registered in https://clinicaltrials.gov/ct2/home (accessed on 20 January 2023), data updated to January 2023.

Recruitment Status	Vaccine	Number of Studies	Phase 1	Phase 1|2	Phase 2	Phase 2|3	Phase 3
Recruiting	Peptides	9	4	NCT05130060NCT04117087NCT02600949NCT04853017	2	NCT04046445NCT03953235	2	NCT04912765NCT05243862	1	NCT05141721	-	
Cell-based	2	1	NCT04147078	-		1	NCT02919644	-		-	
Nucleic acids	1	1	NCT04147078	-		-		-		-	
vector	1	-		-		1	NCT04111172	-		-	
Not yet recruiting	Peptides	3	1	NCT04799431	1	NCT05589597	1	NCT05350501	-		-	
Cell-based	1	1	NCT05235607	-		-		-		-	
Active, not recruiting	Peptides	2	-		2	NCT03639714NCT03761914	-		-		-	
Cell-based	3	2	NCT03730948NCT05238558	1	NCT01885702	-		-		-	
Nucleic acids	1	1	NCT03287427	-		-		-		-	
vector	2	-		1	NCT03563157	1	NCT04491955	-		-	
Terminated	Peptides	5	1	NCT00091286	2	NCT00677612NCT00677287	2	NCT00012246NCT01322815	-		-	
Cell-based	3	1	NCT01952730	-		2	NCT00176761NCT01505166	-		-	
vector	4	2	NCT00088933NCT02714374	-		1	NCT03050814	-		1	NCT01309126
Completed	Peptides	13	6	NCT00641615NCT00006387NCT00128622NCT00020267NCT01522820NCT00019006	4	NCT03391232NCT00019591NCT00785122NCT00861107	3	NCT00773097NCT00019084NCT00019331	-		-	
Cell-based	16	6	NCT00558051NCT01966289NCT00656123NCT01671592NCT00027534NCT00004604	5	NCT03152565NCT00228189NCT00016133NCT02176746NCT01065441	5	NCT02981524NCT02380443NCT00103142NCT01413295NCT00002475	-		-	
Nucleic acids	2	1	NCT03948763	1	NCT01064375	-		-		-	
vector	7	2	NCT01890213NCT00924092	2	NCT00088413NCT00529984	2	NCT04591379NCT00259844	-		1	NCT00427570
Cell-based	4	1	NCT00780988	-		2	NCT02615574NCT03524274	1	NCT01741038	-	
vector	1	-		1		-		-		-	
Unknown status	Peptides	2	2	NCT03689192NCT03552718	-		-		-		-	
Cell-based	4	-		2	NCT00854971NCT00722228	1	NCT01348256	-		1	NCT02503150
vector	2	-		1	NCT00007826	1	NCT00027833	-		-

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
