# Peer review of "Cancer Vaccines: From the State of the Art to the Most Promising Frontiers in the Treatment of Colorectal Cancer"

_pharmaceutics, 2023, doi:10.3390/pharmaceutics15071969_

Round 1
Reviewer 1 Report
The submitted review “Cancer vaccines: from the state of art to the most promising frontiers in treatment of colorectal cancer”, thoroughly describe both the mechanistic and molecular aspects of applying cancer-directed vaccines for managing colorectal cancer (CRC). Additionally, challenges and limitations as well as status of improvements towards forthcoming generations have been highlighted within the submitted manuscript. This manuscript could be considered a valuable guidance for future development and optimization. Suggestions and comments are to be addressed prior publication:
1. In Table 1, authors should provide clear notation regarding which provided CRC vaccines are tumor-specific antigens and which are tumor-associated antigens.
2. “A picture is worth a thousand words”, providing schematic diagrams capable of summarizing lots of information is always considered relevant for any manuscript as well as attractive to the readers for better data comprehension. Therefore, providing a schematic diagram that highlights the mechanistic and molecular aspects of CRC vaccines would be highly relevant.
3. For this submitted review, typical discussion section could be considered inappropriate, other than this, authors could better change it to challenges and future prospections regarding CRC vaccines.
4. Conclusion section should only summarize and highlight key findings, study limitations, and future prospection without redundant results nor introduced tables or figures.
5. In Table 2, authors should the ClinicalTrials.gov identifier (NCT number) for the mentioned studies, as well as appropriate reference to them.
6. Providing a section regarding advents of nanotechnology formulation for different CRC vaccines would also be relevant.
Minor editing of English language required
Author Response
First, we would like to thank the Reviewer for the time dedicated in reviewing our manuscript. We sincerely appreciated the given inputs and the thoughtful critiques, which clearly served to refine and strengthen the original manuscript.
Please find enclosed our revised manuscript entitled “Cancer vaccines: from the state of art to the most promising frontiers in treatment of colorectal cancer” by Eleonora Martinis, Carolina Ricci, Caterina Trevisan, Gaia Tomadini and Silvia Tonon, a review that we submited for your special issue Immunotherapeutic Strategies in Cancer and Chronic Infection.
Here below you will find our point to point revision.
#Reviewer 1
The submitted review “Cancer vaccines: from the state of art to the most promising frontiers in treatment of colorectal cancer”, thoroughly describe both the mechanistic and molecular aspects of applying cancer-directed vaccines for managing colorectal cancer (CRC). Additionally, challenges and limitations as well as status of improvements towards forthcoming generations have been highlighted within the submitted manuscript. This manuscript could be considered a valuable guidance for future development and optimization. Suggestions and comments are to be addressed prior publication:
- In Table 1, authors should provide clear notation regarding which provided CRC vaccines are tumor-specific antigens and which are tumor-associated antigens
We added a column in Table 1 to specify if the antigen is a TAA or a TSA
- “A picture is worth a thousand words”, providing schematic diagrams capable of summarizing lots of information is always considered relevant for any manuscript as well as attractive to the readers for better data comprehension. Therefore, providing a schematic diagram that highlights the mechanistic and molecular aspects of CRC vaccines would be highly relevant
We perfectly agree with reviewer 1. We add two figures, one for peptide, nucleic acid and vector-based vaccine, the other one for DCs- based vaccines.
- For this submitted review, typical discussion section could be considered inappropriate, other than this, authors could better change it to challenges and future prospections regarding CRC vaccines
- Conclusion section should only summarize and highlight key findings, study limitations, and future prospection without redundant results nor introduced tables or figures.
We changed the title of the paragraph in “CRC targeting vaccines under clinical trial” and moved it after the section “CRC targeting-vaccines”. Consequently, we moved table 2 according its citation and changed the title of the last paragraph into “Conclusions and future perspectives”.
- In Table 2, authors should the ClinicalTrials.gov identifier (NCT number) for the mentioned studies, as well as appropriate reference to them
The reference is not available for all studies but we added all the NTC numbers in Table 2
- Providing a section regarding advents of nanotechnology formulation for different CRC vaccines would also be relevant.
We added the section named “Use of nanotechnologies to improve CRC vaccines efficacy”
Reviewer 2 Report
This paper overviewed the promising vaccine treatment of colorectal cancer and proposed some interesting ideas for CRC treatment plan. It is interesting. Some small revisions need to be addressed.
1. HER2, mentioned in “B-cell vaccines: a possible route for cell-based vaccines?” section, is a well-known malignant tumor treatment target, is not present in Table 1. Is there no report about vaccine development against CRC based HER2?
2. Add some mechanism diagrams, such as for “Dendritic cell (DC)-based vaccines for CRC”, or comparison diagrams, such as for “Nucleic acid-based vaccines for CRC” “Viral-, bacterial- and yeast-based vector vaccines for CRC” , would make the article easier to read.
Some minor revisions:
Line 14, When the abbreviations “CRC” first appear, please give the full name.
Line 34, Is there an updated data source after 2020?
Line 78, the word “alternative” appears twice in a sentence. Replace one of them.
Line 100, optimize table format, and a blank row appear in the table
Line 127, Add a reference to this sentence “An example is……”
Line 318, No 1, 2, 3, where did 4 come from?
Line 397, no funding?
Line 414, Incorrect reference format.
Author Response
First, we would like to thank the Reviewer for the time dedicated in reviewing our manuscript. We sincerely appreciated the given inputs and the thoughtful critiques, which clearly served to refine and strengthen the original manuscript.
Please find enclosed our revised manuscript entitled “Cancer vaccines: from the state of art to the most promising frontiers in treatment of colorectal cancer” by Eleonora Martinis, Carolina Ricci, Caterina Trevisan, Gaia Tomadini and Silvia Tonon, a review that we submited for your special issue Immunotherapeutic Strategies in Cancer and Chronic Infection.
Here below you will find our point to point revision.
#Reviewer 2
This paper overviewed the promising vaccine treatment of colorectal cancer and proposed some interesting ideas for CRC treatment plan. It is interesting. Some small revisions need to be addressed.
- HER2, mentioned in “B-cell vaccines: a possible route for cell-based vaccines?” section, is a well-known malignant tumor treatment target, is not present in Table 1. Is there no report about vaccine development against CRC based HER2?
We thank reviewer 2 for this valuable suggestion. We added Her2 among CRC antigens in table 1 and also a brief description of the related vaccines under development in the “peptide-based vaccines for CRC” paragraph.
- Add some mechanism diagrams, such as for “Dendritic cell (DC)-based vaccines for CRC”, or comparison diagrams, such as for “Nucleic acid-based vaccines for CRC” “Viral-, bacterial- and yeast-based vector vaccines for CRC” , would make the article easier to read.
We perfectly agree with reviewer 2. We add two figures, one for peptide, nucleic acid and vector-based vaccine, the other one for DCs- based vaccines.
Line 14, When the abbreviations “CRC” first appear, please give the full name. Done
Line 34, Is there an updated data source after 2020? We retrieved the information from https://gco.iarc.fr/ (Global Cancer Observatory of the WHO), and these are the most updated data
Line 78, the word “alternative” appears twice in a sentence. Replace one of them. Done
Line 100, optimize table format, and a blank row appear in the table Done
Line 127, Add a reference to this sentence “An example is……” Done
Line 318, No 1, 2, 3, where did 4 come from? We deleted all numbers from paragraph title
Line 397, no funding? We added our funding reference
Line 414, Incorrect reference format. Done
Round 2
Reviewer 1 Report
Authors kindly responded to all comments and suggestions.